# The relationship between income poverty and child hospitalisations in New Zealand: Evidence from longitudinal household panel data and Census data

Nichola Shackleton[1], Eileen Li[1], Sheree Gibb[2], Amanda Kvalsvig[2], Michael Baker[2], Andrew Sporle[3], Rebecca Bentley[4], Barry J. Milne[1]*

1 Centre of Methods and Policy Application in the Social Sciences, University of Auckland, Auckland, New Zealand, 2 Department of Public Health, University of Otago, Wellington, New Zealand, 3 Department of Statistics, University of Auckland, Auckland, New Zealand, 4 Centre for Health Equity, Melbourne School of Population and Global Health, University of Melbourne, Melbourne, Australia

* b.milne@auckland.ac.nz

## Abstract

### Background

Very little high quality evidence exists on the causal relationship between income poverty and childhood health. We provide a comprehensive overview of the association between household income poverty and hospitalisations for children.

### Methods

We used New Zealand's Integrated Data Infrastructure (IDI) to link income poverty data from the Survey of Family, Income and Employment (SoFIE; n = 21,759 households) and the 2013 New Zealand Census (n = 523,302 households) to publicly funded hospital records of children aged 0–17 (SoFIE: n = 39,459; Census, n = 986,901). Poverty was defined as equivalised household income below 60% of the median income, calculated both before and after housing costs, and using both self-reported and tax-recorded income.

### Results

Correlations for the association between income poverty and hospitalisation were small (ranging from 0.02 to 0.05) and risk ratios were less than 1.35 for all but the rarest outcome—oral health hospitalisation. Weak or absent associations were apparent across age groups, waves of data collection, cumulative effects, and for estimates generated from fixed effects models and random effect models adjusted for age and ethnicity. Alternative measures of deprivation (area-level deprivation and material deprivation) showed stronger associations with hospitalisations (risk ratios ranged from 1.27–2.55) than income-based poverty measures.

### Conclusion

Income poverty is at best weakly associated with hospitalisation in childhood. Measures of deprivation may have a stronger association. Income measures alone may not be sufficient

**Data Availability Statement:** Integrated Data Infrastructure (IDI) data cannot be shared publicly because of the security and confidentiality

provisions of the Statistics Act 1975 (NZ). Data are held on servers maintained by Statistics New Zealand and are never released to researchers or posted on other sites. As such, the dataset is not owned or available to be distributed by the authors. However, the authors had no special access privileges and other researchers are able to access the data in the same manner as the authors. Researchers who wish to access the data reported in the manuscript for analysis (or access IDI data generally) must submit an application through Statistics New Zealand (https://www.stats.govt.nz/integrated-data/integrated-data-infrastructure#how-apply). Data requests and queries about data access may be sent to access2microdata@stats.govt.nz. If approved, data access is provided through a secure 'Data Lab' environment: a protected virtual environment in secure research facilities on computers that can access the IDI server, but nothing else (i.e. computer hard and soft drives cannot be accessed, and there is no access to the worldwide web). As such, IDI data are never sent to researchers or made available on data archives or via the worldwide web, but instead access is granted to analyse data within the Data Lab environment. Only the results of analyses (e.g., tables, models) can be requested to be released, and these must be confidentialised (e.g., all frequency counts are 'random-rounded' to be divisable by 3). More details on IDI data access is provided in reference 18 of the manuscript (Milne et al., 2019) and here: https://www.stats.govt.nz/integrated-data/integrated-data-infrastructure#data-safe.

**Funding:** This research was funded by a grant (Ref HRC17-250) from the Health Research Council of New Zealand (www.https://hrc.govt.nz) awarded to BM (Principal Investigator) and NS, SG, AK, MB, AS and RB (named investigators). The sponsors played no role in the study design, data collection, analysis, decision to publish, and the preparation of the manuscript.

**Competing interests:** I have read the journal's policy and the authors of this manuscript have the following competing interests: None.

to capture the diversity of household economic circumstances when assessing the poverty-health relationship.

# Introduction

Childhood poverty is increasing in many OECD countries [1–4]. In New Zealand, children have 1.6 times the poverty rate of the overall population—a concentration higher than for any European country [3]. The systemic cycle of poverty perpetuates social and economic differences between ethnic groups, and traps families in poverty across generations. To break this cycle, policy makers need evidence on the most effective means of reducing the impact of poverty on a range of outcomes, including the health of children in families experiencing poverty.

Rising rates of child poverty in New Zealand have been linked to increasing rates in hospital admissions for asthma, pneumonia, rheumatic fever and serious skin infections [5]. Furthermore children living in poverty in New Zealand have a 40% higher risk of dying during childhood [6], are sick three times as often [7], are more than twice as likely to be admitted to hospital for an acute infectious disease [8], miss out on more days of school due to ill health [9], and are more likely to have poor health outcomes in adulthood, including cardiovascular disease, dental decay, lowered longevity and mental illness [5]. As well as the association of poverty with child and subsequent adult health, children living in poverty have poorer education, social and economic outcomes in adulthood [3].

A number of theories have been posited to explain the relationship between poverty and child health. One theory–the 'economic investment' theory–suggests that more affluent parents are better able to 'invest' more in their children's development (e.g., through investments in nutrition, clothing, learning resources, housing, and healthcare) and this leads to better outcomes [10]. Affluence can also overcome issues of access to health care, particularly if cost is a barrier, but even when it is not: e.g., New Zealand has universal health care with low co-payments in primary care, yet the odds of unmet need for primary health care is 1.4 times greater among those living in the most versus least deprived neighbourhoods [11]. Another theory–the 'family stress' theory–suggests that living in poverty increases parental stress levels, and this stress hinders parents' ability to provide quality care, which has negative impacts upon children's outcomes [12, 13]. Other factors, such as health behaviours and health literacy may also play a role [14, 15]. Note that these factors are not mutually exclusive: the effect of poverty on children's health could operate simultaneously through many pathways.

Although there is observational evidence for an association between poverty and a range of child health outcomes, very little high quality evidence exists on the causal relationship between poverty and childhood health [10, 16]. This is a major barrier to developing policy, because effective intervention requires a clear understanding of causal effects, unconfounded by other factors. For example, while the cost to society of 'doing nothing' has been estimated to exceed the likely cost of addressing poverty [17], policy makers need to know whether giving money, or money in kind, to low income families will in fact improve the health of children.

Cooper and Stewart (2013; 2017) [10, 16] synthesised evidence from studies that attempt to isolate the causal effect of low income on child outcomes, either through quasi-experimental designs (e.g., programmes that increased incomes) or by statistical isolating the impact of low-income by analysing within household differences or within-individual change. In their original review, eight studies investigating physical health outcomes showed mixed evidence of a relationship [10]. This was increased to 17 studies in an updated review (15 studies using

quasi-experimental designs and two using observational designs) [16]. They found that income was important for child health, but this varied for the type of health outcomes assessed [16]. There was evidence pointing to significant effects of exogenous changes in income (i.e., through payments or tax credits) on birthweight and other neonatal outcomes, as well as evidence that differences in household income experienced by children within families explained birthweight differences between children. However, there was mixed evidence for the impact of exogenous changes in income on obesity and general health in later childhood, and no evidence (from three studies) for an effect of exogenous changes in income on asthma, wheezing and other respiratory diseases [16].

This study will assess associations between child poverty and hospitalisation outcomes using data from the Statistics New Zealand Integrated Data Infrastructure (IDI), a collection of de-identified administrative datasets that have been linked at the person-level for the whole New Zealand population [18]. This study contributes to the existing literature in the following ways. First, it represents an analysis of associations between poverty / low income and a range of hospitalisations thought to be poverty sensitive, specifically: respiratory conditions [19, 20], infectious diseases [8], otitis media [18], oral health [21, 22], and preventable admissions [23, 24]. Second, we assess poverty in two samples: a longitudinal household panel survey: the Survey of Family, Income and Employment [25]; and whole population data from the 2013 New Zealand census; and we measure income from two sources: self-report and administrative tax records. Third, there have been debates regarding 'sensitive periods' for poverty experience, with some emphasising the importance of early childhood [26], while others pointing to the importance of adolescence [27]. We make use of our multi-age cohorts to assess whether associations between poverty and health are age-sensitive. Fourth, we make use of longitudinal data to assess whether duration of time in poverty has an impact on hospitalisation outcomes, as has been suggested for other health outcomes [27]. Fifth, our use of longitudinal data allows us to analyse within-individual change over an extended period–eight annual data collection waves–to estimate if associations are likely to be causal. Specifically, fixed effects regression analyses focus on within-individual change and control all 'fixed' differences (i.e., unchanging–or time invariant–confounding) between individuals that might otherwise explain associations between poverty and health [16, 28].

We answer four research questions:

1. Is the risk of hospitalisation greater for children in income poverty?

2. Does the risk of hospitalisation vary by the age of the child?

3. Does the risk of hospitalisation vary by length of exposure to poverty?

4. Is there evidence of a causal relationship between income poverty and children's hospitalisations in New Zealand?

## Materials and methods

### Participants

We used data on children from two sources: (i) a longitudinal household panel survey: the Survey of Family, Income and Employment (SoFIE) [25]; and (ii) a cross-sectional snapshot of the entire population: the 2013 New Zealand census. Data were sourced from the IDI, (https://www.stats.govt.nz/integrated-data/integrated-data-infrastructure/) and were accessed through Statistics New Zealand, which manages the privacy, security, and confidentiality of administrative and survey data in the IDI. Our analytic sample of children included all respondents aged between 0 to 14, as well as respondents aged 15 to 17 years classified as 'dependent'.

This study received ethics approval from the University of Auckland Human Ethics Committee (019618). Access to the anonymised data used in this study, including both SoFIE and Census data, was provided by Statistics NZ under the security and confidentiality provisions of the Statistics Act 1975.

**SoFIE.** The SoFIE longitudinal survey allowed us to follow families and their children over time so that we could examine how changes in poverty status affected children's hospitalisations. SoFIE participants were sampled from the usually resident population living in permanent, private dwellings on the main islands of New Zealand, and were followed for up to eight annual waves (September 2002—September 2010). Information about children within households and families was recorded by adult respondents. Written consent was obtained from 79% adult participants to link to administrative health records [29]. Children were excluded from analyses if they had a consent-refusing parent at any wave, or if they refused consent themselves when aged 15 or over.

**Census.** The 2013 Census took place on 5 March 2013 and counted 97.6 percent of New Zealand residents in the country. n = 986,901 children were included in our sample.

## Measures

**Income from self-report.** *SoFIE*. Income was calculated for the 12 months preceding the enumeration date. 'Household self-reported income before housing costs' (BHC) is the sum of the annual gross personal income for all individuals in the child's household. Annual gross personal income was derived by adding together: Employee earnings, government transfer income, income from self-employment, interest from bank accounts, income from other investments, income from private superannuation and pension schemes, other income received as regular payments, and other irregular income.

'Household self-reported disposable income' (disposable) was calculated as BHC income minus (i) housing costs (including rent, mortgage, and rates, land rates, body corporate fees, and water rates); and (ii) the calculated tax paid for self-reported income (given tax bands for each year).

*Census.* The census question asks: "From all the sources of income . . .. what will the total income be: that you yourself got before tax or anything was taken out of it, in the 12 months that will end on 31 March 2013". The response is collected in bands (loss, zero income, $1-$5,000, $5,001-$10,000, $10,001-$15,000, $15,001-$20,000, $20,001-$25,000, $30,001-$35,000, $40,001-$50,000, $50,001-$60,000, $60,001-$70,000, $70,001-$100,000, $100,001-$150,000, $150,001 or more). The median point of each band was assigned to individuals and these were summed for all individuals in the child's household to derive 'household self-reported income before housing costs' (BHC). It was not possible to calculate disposable income using Census records.

**Income from tax records.** For both the SoFIE and Census cohorts, income data from tax records was obtained for members of the child's household from the New Zealand Inland Revenue Department (IRD). IRD income includes monthly income taxed at source and additional income which is captured annually in April via tax returns. Monthly income includes: Wages and Salaries, Withholding payments, tier 1 Benefit payments, Accident Compensation payments, Pension payments, Paid Parental Leave payments, Student Allowance payments, Company director / shareholder receiving "pay as you earn" (PAYE) or tax deducted income, Partner receiving PAYE or tax deducted income, Sole trader receiving PAYE or tax deducted income. Additional income includes: partnership income, director / shareholder income, sole trader income, and rental income. The entitlement amount from tier 2 benefits, including the accommodation supplement, disability allowance, and child disability allowance was added to IRD income.

For monthly income the annual total income was calculated by summing the monthly income from the 12 months prior to the enumeration date. For additional income we estimated income from the 12 months prior to the enumeration date by summing the required proportions of income from the current and previous tax year. IRD records were missing and excluded from analyses for children who resided in households where no adult had an IRD record (<3% for SoFIE; 5.5% for Census).

**Income poverty calculation.**   Income was trimmed to have a lower bound of $1000 and upper bound of the 99th percentile (this differed for each SoFIE wave and for the Census cohort). The lower bound was set to allow log transformations of income, which takes into account the proportional value of increases in income. Trimmed household income was equivalised using the New Zealand-specific Jensen equivalence scale (1988) [30].

Following national and international standards [3], we define children with equivalised household income below 60% of the median income (using both the BHC and disposable measures) as living in poverty. Sensitivity checks were conducted with poverty defined as 50% below the median income but the substantive findings do not differ to those presented here.

**Hospitalisation.**   Hospitalisations were derived from the Ministry of Health's National Minimum Dataset. This includes data on all admissions to New Zealand's public hospitals since 1988. Hospitalisations were recorded for the 12-month period following the enumeration date for the following conditions: infectious diseases, respiratory conditions, preventable admissions, otitis media, and oral health. We collapsed counts of hospitalisation in the 12 month period into binary indicators (yes/no) due to the small number of children who were hospitalised more than once for a condition per 12 month period. We also generated an overall indicator of hospitalisation for any condition. A full list of codes used to identify hospitalisations for each condition is shown in S1 Appendix.

**Deprivation.**   Two alternative measures of poverty were assessed.

Area level deprivation was estimated using the New Zealand Index of Deprivation–the NZDep2001 for SoFIE [31] and the NZDep2013 for the census [32]–based on the deprivation characteristics of 'meshblocks' (small areas with a typical population of 60–110 people). The New Zealand Index of Deprivation combines census data relating to income, employment, qualifications, communication, support, living space, transport and home ownership into a single measure of relative socio-economic deprivation [31]. Meshblocks were ranked from least to most deprived and assigned to quintiles.

Material deprivation was assessed using the New Zealand Index of Socioeconomic Deprivation for Individuals (NZiDep) at waves 3, 5 and 7 of SoFIE (material deprivation was not available in Census data). The NZiDep index consists of eight questions covering: means-tested benefits, unemployment, receiving help from a community organisation, assistance obtaining food, wearing worn-out shoes, buying cheap food, going without fresh fruit and vegetables, and feeling cold to save on heating costs [33]. It is scored on a five-category scale (from no deprivation characteristics to five or more deprivation characteristics) [33]. We used the highest parental score to assign deprivation status to children. Where parental information was missing, information provided by grandparents, aunts/uncles, or other adults in the household was used.

*Age* (in years), *Gender* (male/female) and *Ethnicity* were included in models as potential confounders. Ethnicity was taken from the 'source ranked ethnicity' variable in IDI, which collates and ranks ethnicity classifications from different data sources (census records are ranked highest, followed by birth records, followed by other sources). Six binary variables for the following ethnic groups were created: European; Māori; Pacific; Asian; Middle Eastern, Latin American or African; and Other. Individuals could belong to one or more of these ethnic groups, in line with the Statistical Standard for reporting ethnicity [34].

## Statistical analysis

Descriptively, we present poverty rates and prevalence of each hospitalisation for both samples on aggregate, and by wave for the SoFIE sample.

Our first two research questions used both SoFIE and Census data. To answer the first research question, "Is the risk of hospitalisation greater for children in income poverty?", we computed the unadjusted associations between poverty and hospitalisation outcomes by running cross-tabulations and calculating unadjusted relative risks. SoFIE analyses were undertaken for each wave separately. Poverty was assessed using three measures: (i) self-reported gross income before housing costs (BHC–self report); (iii) tax-recorded gross income before housing costs (BHC–tax records); and (iii) self-reported disposable income (Disposable–self report). We used non-parametric local polynomial smoothing [35] to establish the functional form for the unadjusted association between income (logged) and the hospitalisation outcomes. To answer the second research question, 'Does the risk of hospitalisation vary by the age of the child?', we investigated associations stratified by age group.

Our third and fourth research questions required longitudinal data so were restricted to the SoFIE cohort only. To answer the third question, 'Does the risk of hospitalization vary by length of exposure to poverty?', we assessed associations between accumulation of poverty over two consecutive waves and hospitalisations. To answer the fourth question, 'Is there evidence of a causal relationship between income poverty and children's hospitalisations in New Zealand?', we tested the association between income poverty/income and hospitalisations adjusted for other factors using regression models across all eight waves of data, employing both fixed effect models and random effects (random intercepts). Fixed effect models use only the 'within' person variance–change in predictors and outcomes over time for an individual, whereas random effects model use both the 'within' and 'between' person variance–differences in the relationship between individual means in the predictors and outcomes–to produce estimates. Fixed effect models allow us to better account for unmeasured confounding, as time invariant (stable) characteristics of individuals are controlled. However, fixed effect models are restricted to the subsample who have changes in recorded hospitalisations per year. In contrast, random effect models allow us to investigate associations between income and health outcomes for the entire sample, so can be used to make inferences to the population, under certain assumptions. Note, poverty status was variable across waves: 28.1% of children changed poverty status at least once across the eight waves.

We conducted a sensitivity analysis using two alternative measures of poverty (area-level deprivation and material deprivation) to test whether these showed the same pattern of associations with hospitalisation outcomes. Area level deprivation could be measured in both the SoFIE and Census cohorts, but material deprivation could only be assessed in the SoFIE cohort, and only at three waves (3, 5 and 7).

We conducted two additional sensitivity analyses using the SoFIE sample. First, to assess whether attrition impacted associations, we investigated associations between poverty and hospitalisations by wave. Second, to account for the error associated with single measures of income, we created rolling averages of income over two waves in SoFIE and assessed the association between time-averaged income and hospitalisations.

Data were analysed using Stata version 15 (StataCorp LLC) [36].

## Results

Descriptive results are shown in Table 1. The SoFIE cohort (2002–2010) included 39,459 person-waves, comprising 9,276 individuals across 4.25 waves on average. For this cohort, the median self-reported equivalised household income before housing costs was $44,080 (IQR:

**Table 1. Characteristics of the SOFIE and Census samples.**

| | SoFIE | | | | | | | | | Census |
|---|---|---|---|---|---|---|---|---|---|---|
| | Overall | Wave1 | Wave2 | Wave3 | Wave4 | Wave5 | Wave6 | Wave7 | Wave8 | |
| number of children | 39,459[a] | 6,258 | 5,646 | 5,127 | 4,833 | 4,683 | 4,437 | 4,278 | 4,197 | 986,901 |
| number of households | 21,759 | 3,423 | 3,090 | 2,823 | 2,670 | 2,568 | 2,472 | 2,394 | 2,316 | 523,302 |
| Age, mean (SD) | 8.56 (5.03) | 8.31 (5.01) | 8.45 (5.03) | 8.59 (5.00) | 8.62 (5.01) | 8.62 (5.02) | 8.60 (5.02) | 8.64 (5.06) | 8.73 (5.11) | 8.25 (5.11) |
| Age group, % | | | | | | | | | | |
| 0–4 yrs | 26.09 | 27.61 | 26.83 | 25.45 | 25.39 | 25.37 | 25.76 | 25.67 | 26.16 | 29.03 |
| 5–10 yrs | 34.86 | 35.19 | 35.33 | 35.87 | 35.63 | 34.85 | 34.55 | 33.94 | 32.81 | 33.94 |
| 11–17 yrs | 39.05 | 37.15 | 37.83 | 38.68 | 38.92 | 39.85 | 39.62 | 40.39 | 41.10 | 37.03 |
| Gender, % | | | | | | | | | | |
| Male | 50.70 | 50.62 | 50.85 | 51.02 | 50.47 | 50.67 | 50.44 | 50.70 | 50.75 | 50.99 |
| Female | 49.31 | 49.38 | 49.10 | 48.98 | 49.53 | 49.39 | 49.56 | 49.37 | 49.32 | 49.01 |
| Ethnicity[b], % | | | | | | | | | | |
| European | 79.15 | 72.91 | 75.93 | 78.64 | 80.32 | 81.29 | 81.54 | 82.61 | 83.63 | 67.76 |
| Maori | 26.34 | 28.28 | 26.94 | 26.92 | 25.70 | 25.94 | 25.56 | 25.32 | 24.80 | 23.01 |
| Pacific | 9.99 | 12.27 | 11.11 | 9.42 | 9.12 | 9.10 | 9.26 | 9.47 | 9.08 | 12.18 |
| Asian | 5.55 | 6.23 | 6.06 | 5.68 | 5.40 | 5.32 | 5.54 | 4.91 | 4.79 | 11.60 |
| MELAA | 2.33 | 3.40 | 3.24 | 2.98 | 2.67 | 2.18 | 2.03 | 1.54 | 1.36 | 1.42 |
| Other | 1.40 | 1.39 | 1.43 | 1.40 | 1.49 | 1.28 | 1.28 | 1.47 | 1.50 | 1.43 |
| Hospitalisations, % | | | | | | | | | | |
| Otitis Media | 0.60 | 0.67 | 0.96 | 0.59 | 0.50 | 0.58 | 0.54 | 0.49 | 0.50 | 0.42 |
| Oral Health | 0.47 | 0.53 | 0.43 | 0.53 | 0.56 | 0.51 | 0.47 | 0.28 | 0.43 | 0.56 |
| Infectious | 3.04 | 3.74 | 3.19 | 2.75 | 2.92 | 2.95 | 3.11 | 2.81 | 2.79 | 2.91 |
| Respiratory | 0.95 | 1.10 | 0.90 | 0.94 | 0.87 | 0.90 | 1.08 | 0.91 | 0.93 | 0.92 |
| Preventable | 3.09 | 3.88 | 3.35 | 2.87 | 2.92 | 2.88 | 2.97 | 2.73 | 2.79 | 2.92 |
| All hospitalisations | 7.25 | 8.15 | 7.33 | 6.67 | 6.89 | 6.73 | 7.30 | 7.22 | 7.51 | 7.16 |
| Income poverty, % | | | | | | | | | | |
| BHC[c] –self report | 30.77 | 33.32 | 31.77 | 30.08 | 29.17 | 30.88 | 29.89 | 30.08 | 29.95 | 29.0 |
| BHC[c] –tax records | 30.64 | 31.44 | 30.91 | 30.15 | 30.28 | 29.36 | 30.68 | 31.67 | 30.47 | 32.9 |
| Disposable[d] –self report | 35.96 | 37.34 | 37.35 | 34.70 | 33.46 | 35.43 | 35.50 | 35.90 | 36.53 | – |

[a] This represents the sum of children participating across the eight waves, which we refer to throughout the paper 'person-waves', i.e., the number of children*the number of waves. Overall there are 9276 unique children in the SoFIE sample who participate for an average of 4.25 waves. The n for each wave represents individual children, e.g., in wave 1 there are 6,258 unique children.

[b] Ethnic groups are not mutually exclusive, a child can be classified as belonging to multiple ethnicities.

[c] BHC = Before Housing Costs. Based on 60% of the median income after equivalisation.

[d] Disposable. Based on 60% of the median of income, less housing costs and tax, and then equivalised.

$26,313 - $68,631) with 30.8% children classified as in poverty (BHC–self report). The median tax-recorded equivalised household income before housing cost was $37,290 (IQR: $21,212 - $59,227), with 30.6% children classified as in poverty (BHC–tax records). The median self-reported disposable income was $25,361 (IQR: $13,333 - $41,344), with 36.0% classified as in poverty (Disposable–self report). There was a very strong correlation between self-reported BHC income and disposable income (spearman's rho($\rho$) = 0.95), and a lower correlation between self-reported and tax-recorded BHC income ($\rho$ = 0.75).

The Census cohort (2013) included 986,901 children from 523,302 households. For this cohort, the median self-reported equivalised household income before housing costs was $51,145 (IQR: $31,930 - $81,458) with 29.0% children classified as in poverty. The median tax-

recorded equivalised household income before housing cost was $45,356 (IQR: $24,609 - $73,339), with 32.9% children classified as in poverty. The correlation between self-reported and tax-recorded BHC income was $\rho = 0.80$.

## Is the risk of hospitalisation greater for children in income poverty?

To quantify the association between income poverty and hospitalisation, we compared unadjusted relative risks of hospitalisation for those in poverty compared to those not in poverty (Table 2). For self-reported BHC poverty, relative risks ranged from 1.07 to 1.28 in SoFIE and 1.16 to 1.77 in the Census. For tax-recorded BHC poverty, relative risks ranged from 0.95 to 1.43 in SoFIE and 1.04 to 1.56 in the Census. Estimates using self-report income were slightly larger in the Census than in SoFIE (though all but one Census estimate was within confidence limits of the equivalent SoFIE estimate). Estimates using tax-recoded income were very similar between Census and SoFIE. Relative risk estimates for disposable income poverty ranged from 1.13–1.43. All associations can be considered weak: tetrachoric correlations for the associations in Table 2 ranged from r = 0.02 to r = 0.05.

Local polynomial smooths (see S2 Appendix) provide evidence for a modest non-linear relationship between income and hospitalisations, whereby there is little difference in the hospitalisation rate for those between the 1st and 60th percentile of the income distribution (i.e., similar risk among those of low-moderate income), and a decrease from the 60th percentile onwards (i.e., diminishing risk with increasing income above the 60th percentile).

**Table 2. Association (unadjusted) between income poverty and hospitalisations using self-reported and tax recorded income for the SoFIE and Census samples.**

| | SoFIE | | | Census | | |
|---|---|---|---|---|---|---|
| | Not in poverty (%) | In poverty (%) | RR | Not in poverty (%) | In poverty (%) | RR |
| **BHC–self report[a]** | | | | | | |
| Otitis Media | 0.56 | 0.72 | 1.28 (0.98;1.66) | 0.41 | 0.51 | 1.23 (1.15, 1.32) |
| Oral Health | 0.45 | 0.54 | 1.18 (0.88;1.60) | 0.45 | 0.80 | 1.77 (1.67, 1.87) |
| Infectious | 2.90 | 3.43 | 1.18 (1.04;1.32) | 2.73 | 3.48 | 1.27 (1.24, 1.31) |
| Respiratory | 0.91 | 1.09 | 1.20 (0.97;1.49) | 0.85 | 1.13 | 1.33 (1.27, 1.39) |
| Preventable | 2.97 | 3.43 | 1.16 (1.03;1.30) | 2.69 | 3.56 | 1.33 (1.29, 1.36) |
| All hospitalisations | 7.13 | 7.66 | 1.07 (0.99;1.16) | 7.02 | 8.16 | 1.16 (1.14, 1.18) |
| **BHC–tax records[a]** | | | | | | |
| Otitis Media | 0.62 | 0.59 | 0.95 (0.72;1.26) | 0.43 | 0.45 | 1.04 (0.97, 1.11) |
| Oral Health | 0.42 | 0.61 | 1.43 (1.06;1.92) | 0.49 | 0.77 | 1.56 (1.48, 1.65) |
| Infectious | 2.99 | 3.30 | 1.11 (0.98;1.25) | 2.86 | 3.35 | 1.17 (1.15, 1.20) |
| Respiratory | 0.91 | 1.12 | 1.22 (0.98;1.50) | 0.89 | 1.11 | 1.25 (1.20, 1.31) |
| Preventable | 2.99 | 3.45 | 1.16 (1.03;1.30) | 2.83 | 3.44 | 1.22 (1.19, 1.25) |
| All hospitalisations | 7.32 | 7.31 | 1.00 (0.93;1.08) | 7.22 | 7.90 | 1.09 (1.08, 1.11) |
| **Disposable–self report[b]** | | | | | | |
| Otitis Media | 0.52 | 0.76 | 1.43 (1.11;1.85) | | | |
| Oral Health | 0.42 | 0.55 | 1.28 (0.96;1.72) | | | |
| Infectious | 2.83 | 3.47 | 1.23 (1.10;1.39) | | | |
| Respiratory | 0.89 | 1.10 | 1.23 (1.00;1.51) | | | |
| Preventable | 2.88 | 3.53 | 1.23 (1.10;1.40) | | | |
| All hospitalisations | 6.94 | 7.89 | 1.13 (1.06;1.22) | | | |

[a] BHC = Before Housing Costs. Based on 60% of the median income after equivalisation.

[b] Based on 60% of the median of income, less housing costs and tax, and then equivalised.

**Table 3. Association between income poverty and hospitalisations stratified by age groups for the SoFIE and Census samples.**

|  | Otitis Media | Oral Health | Infectious | Respiratory | Preventable | Any admission |
|---|---|---|---|---|---|---|
| **SoFIE** |  |  |  |  |  |  |
| **BHC–self report** |  |  |  |  |  |  |
| 0–4 yrs | 1.14 (0.81;1.61) | 0.87 (0.54;1.40) | 1.12 (0.97;1.31) | 1.25 (0.98;1.57) | 1.11 (0.96;1.28) | 1.09 (0.97;1.22) |
| 5–10 yrs | 1.40 (0.89;2.19) | 1.22 (0.79;1.89) | 1.21 (0.97;1.54) | 0.76 (0.41;1.39) | 1.12 (0.89;1.40) | 1.05 (0.91;1.21) |
| 11–17 yrs | 1.10 (0.38;3.17) | 2.40 (0.96;6.11) | 1.09 (0.81;1.46) | 1.09 (0.49;2.40) | 1.18 (0.84;1.68) | 1.04 (0.90;1.20) |
| BHC–tax records |  |  |  |  |  |  |
| 0–4 yrs | 0.71 (0.48;1.03) | 0.96 (0.60;1.53) | 0.96 (0.82;1.12) | 1.16 (0.91;1.47) | 1.02 (0.88;1.18) | 0.98 (0.87;1.10) |
| 5–10 yrs | 1.19 (0.75;1.91) | 1.62 (1.06;2.50) | 1.30 (1.03;1.64) | 1.18 (0.67;2.09) | 1.19 (0.94;1.50) | 1.02 (0.89;1.18) |
| 11–17 yrs | 1.91 (0.71;5.12) | 3.16 (1.18;8.47) | 1.23 (0.92;1.64) | 1.23 (0.57;2.62) | 1.52 (1.09;2.14) | 0.97 (0.83;1.12) |
| Disposable–self report |  |  |  |  |  |  |
| 0–4 yrs | 1.21 (0.87;1.69) | 0.95 (0.61;1.49) | 1.17 (1.01;1.35) | 1.24 (0.98;1.55) | 1.13 (0.98;1.29) | 1.12 (1.00;1.25) |
| 5–10 yrs | 1.52 (0.97;2.36) | 1.24 (0.81;1.89) | 1.31 (1.05;1.66) | 0.80 (0.45;1.42) | 1.19 (0.96;1.49) | 1.09 (0.96;1.25) |
| 11–17 yrs | 1.54 (0.57;1.41) | 3.10 (1.20;8.03) | 0.97 (0.73;1.30) | 0.89 (0.41;1.96) | 1.17 (0.83;1.64) | 1.12 (0.97;1.28) |
| **Census** |  |  |  |  |  |  |
| BHC–self report |  |  |  |  |  |  |
| 0–4 yrs | 1.00 (0.91;1.09) | 1.94 (1.80;2.11) | 1.22 (1.18;1.26) | 1.26 (1.19;1.33) | 1.23 (1.19;1.27) | 1.13 (1.11;1.16) |
| 5–10 yrs | 1.57 (1.39;1.77) | 1.44 (1.32;1.58) | 1.28 (1.21;1.34) | 1.27 (1.14;1.41) | 1.35 (1.28;1.42) | 1.16 (1.12;1.19) |
| 11–17 yrs | 1.56 (1.18;2.06) | 1.61 (1.23;2.10) | 1.28 (1.18;1.37) | 1.69 (1.42;2.01) | 1.49 (1.37;1.62) | 1.16 (1.12;1.20) |
| BHC–tax records |  |  |  |  |  |  |
| 0–4 yrs | 0.87 (0.80;0.94) | 1.69 (1.57;1.82) | 1.13 (1.09;1.16) | 1.19 (1.14;1.26) | 1.15 (1.12;1.18) | 1.07 (1.05;1.09) |
| 5–10 yrs | 1.22 (1.09;1.36) | 1.37 (1.26;1.48) | 1.19 (1.14;1.25) | 1.24 (1.13;1.37) | 1.25 (1.20;1.31) | 1.09 (1.06;1.18) |
| 11–17 yrs | 1.72 (1.36;2.17) | 1.28 (1.00;1.65) | 1.19 (1.11;1.27) | 1.49 (1.29;1.73) | 1.34 (1.25;1.45) | 1.11 (1.08;1.15) |

## Does the risk of hospitalisation vary by the age of the child?

In SoFIE, there was no evidence for differential associations between income poverty and hospitalisation rates in different age groups, as evidence by a lack of consistent patterning in the relative risk estimates by age group and overlapping confidence intervals for all three age groups (Table 3). In the Census there was also no persistent pattern, though associations were stronger for otitis media hospitalisations in over 5s, and in respiratory conditions in over 10s.

## Does the risk of hospitalisation vary by length of exposure to poverty?

To assess the association between persistent poverty and hospitalisations, we restricted our analysis to the longitudinal SoFIE sample and assessed hospitalisation rates by number of waves in poverty across two consecutive waves (there were n = 28,863 person-waves in which children were included in at least two consecutive waves). Children who were in poverty for one of the two of the previous waves tended to have the highest rates of hospitalisations across outcomes (Table 4). That is, persistence (two consecutive waves) of poverty was not associated with an increasing likelihood of hospitalisation.

## Is there evidence of a causal relationship between income poverty and children's hospitalisations in New Zealand?

The fixed effect models for BHC income poverty (Table 5) suggested that transitioning into poverty was associated with lower odds of being hospitalised in the following 12-month period for oral health (0.49(0.29;0.83)) and preventable admissions (0.77(0.62;0.97)). All other odds ratios had confidence intervals overlapping with 1, suggesting no evidence for an association

**Table 4. Association (unadjusted) between income poverty across two consecutive waves and hospitalisations using self-reported and tax recorded income for the SoFIE sample.**

| | Otitis Media | Oral Health | Infectious | Respiratory | Preventable | Any admission |
|---|---|---|---|---|---|---|
| **BHC–self report** | | | | | | |
| In poverty | % | % | % | % | % | % |
| 0 waves (63.0%) | 0.50 | 0.43 | 2.33 | 0.69 | 2.34 | 6.50 |
| 1 wave (15.0%) | 0.83 | 0.55 | 2.98 | 0.49 | 2.77 | 7.07 |
| 2 waves (22.0%) | 0.66 | 0.61 | 2.69 | 0.66 | 2.78 | 6.51 |
| Pearson Chi-squared | $\chi^2(2) = 8.02, p<0.05$ | $\chi^2(2) = 4.24, p>0.05$ | $\chi^2(2) = 7.79, p<0.05$ | $\chi^2(2) = 1.49, p>0.05$ | $\chi^2(2) = 5.32, p>0.05$ | $\chi^2(2) = 2.05, p>0.05$ |
| **BHC–tax records** | | | | | | |
| In poverty | % | % | % | % | % | % |
| 0 waves (65.0%) | 0.57 | 0.41 | 2.42 | 0.63 | 2.34 | 6.67 |
| 1 wave (11.2%) | 0.75 | 0.66 | 3.30 | 0.75 | 3.49 | 7.45 |
| 2 waves (23.7%) | 0.49 | 0.62 | 2.50 | 0.71 | 2.54 | 6.28 |
| Pearson Chi-squared | $\chi^2(2) = 1.13, p>0.05$ | $\chi^2(2) = 6.42, p<0.05$ | $\chi^2(2) = 7.32, p<0.01$ | $\chi^2(2) = 1.07, p>0.05$ | $\chi^2(2) = 14.55, p<0.01$ | $\chi^2(2) = 4.58, p>0.05$ |
| **Disposable–self report** | | | | | | |
| In poverty | % | % | % | % | % | % |
| 0 waves (56.7%) | 0.46 | 0.44 | 2.25 | 0.64 | 2.20 | 6.32 |
| 1 wave (16.7%) | 0.87 | 0.37 | 3.04 | 0.74 | 3.22 | 7.38 |
| 2 waves (26.5%) | 0.59 | 0.67 | 2.79 | 0.63 | 2.79 | 6.75 |
| Pearson Chi-squared | $\chi^2(2) = 14.03, p<0.01$ | $\chi^2(2) = 5.61, p>0.05$ | $\chi^2(2) = 11.62, p<0.01$ | $\chi^2(2) = 1.08, p>0.05$ | $\chi^2(2) = 17.81, p<0.05$ | $\chi^2(2) = 7.22, p<0.05$ |

Note: N = 28863 person-waves were included in the analysis for BHC income poverty, N = 28857 person-waves for disposable income and N = 28839 person-waves for IRD income poverty.

between changes in the log of BHC income and hospitalisations. We tested for one-year lags for income poverty but the results were similar to those presented here.

The random intercepts models also provide no evidence for associations between BHC income poverty, or the log of BHC income with hospitalisation outcomes. As the log of income did not have a linear association with hospitalisation outcomes, we also tested quintiles of income. Analyses of quintiles showed some evidence of associations for 'oral health', infectious disease admissions', 'preventable admissions' and 'any admission', for which the poorest four quintiles have similar risk, which was elevated compared to the richest quintile.

**Sensitivity analyses.** We tested associations between non-income measures of poverty and hospitalisation outcomes. The first column of Table 6 shows that both non-income measures of poverty–area deprivation and material deprivation–were associated with income-poverty. For example, in SoFIE, 14.5% of those residing in the least deprived 20% of areas were classified as living in BHC income poverty, compared to 53.9% of those living in the most deprived 20% of areas; whereas in the Census, the prevalence of poverty in the least and most deprived areas was 7.6% and 47.6%, respectively. Also, 16.9% of those reporting no material deprivation in SoFIE were classified as living in BHC income poverty, compared to 75.4% of those reporting 5 or more indicators of material deprivation. BHC income was moderately and negatively correlated with area-level deprivation (p = -.39) and material deprivation (p = -.44).

The remaining columns of Table 6 show associations between area-level deprivation and hospitalisation outcomes for SoFIE (upper panel) and for the Census (middle panel); and between material deprivation and hospitalisation outcomes for SoFIE (lower panel). Greater levels of area-level deprivation were associated with greater risk of hospitalisation across outcomes, with the risk of children in quintile 5 having hospitalisation outcomes being 1.27–2.20 times that of children in quintile 1 for SoFIE; and 1.36–2.32 times that of children in quintile 1

**Table 5. Longitudinal associations association between income (poverty) and hospitalisations in the SOFIE sample (n = 38,919 observations for n = 9,216 children).** Panels show odd ratios (95% confidence intervals).

| | Otitis Media | Oral Health | Infectious | Respiratory | Preventable | Any admission |
|---|---|---|---|---|---|---|
| **Fixed effect models** | | | | | | |
| BHC income poverty | 0.75(0.46;1.21) | 0.49(0.29;0.83) | 0.82(0.66;1.03) | 1.26(0.80;2.00) | 0.77(0.62;0.97) | 0.89(0.76;1.03) |
| Log of BHC income | 1.44(1.02;2.03) | 1.28(0.87;1.90) | 1.10(0.96;1.26) | 0.91(0.70;1.20) | 1.10(0.96;1.27) | 1.00(0.92;1.09) |
| Number of children | 162 | 162 | 873 | 273 | 837 | 1857 |
| Number of observations | 1029 | 1023 | 5103 | 1470 | 4905 | 11007 |
| **Random intercepts models** | | | | | | |
| BHC income poverty | 1.07(0.76;1.52) | 0.96(0.68;1.34) | 1.04(0.90;1.20) | 1.02(0.78;1.34) | 0.99(0.86;1.15) | 1.01(0.92;1.12) |
| Log of BHC income | 0.91(0.76;1.08) | 0.98(0.82;1.17) | 0.96(0.89;1.03) | 1.01(0.88;1.15) | 0.96(0.89;1.03) | 0.95(0.91;1.00) |
| BHC income quintiles | | | | | | |
| 1 (lowest 20%) | 1.27(0.71;2.27) | 1.59(0.87;2.89) | 1.26(1.00;1.59) | 0.99(0.65;1.51) | 1.33(1.04;1.69) | 1.19(1.02;1.39) |
| 2 | 1.48(0.85;2.60) | 1.88(1.06;3.35) | 1.29(1.03;1.61) | 0.97(0.64;1.48) | 1.31(1.03;1.66) | 1.29(1.11;1.50) |
| 3 | 1.47(0.85;2.55) | 1.88(1.06;3.35) | 1.37(1.10;1.70) | 1.02(0.68;1.53) | 1.43(1.13;1.81) | 1.31(1.13;1.51) |
| 4 | 0.91(0.51;1.64) | 1.72(0.95;3.09) | 1.12(0.89;1.41) | 0.97(0.64;1.47) | 1.24(0.98;1.58) | 1.12(0.96;1.30) |
| 5 (highest 20%) | Ref | Ref | Ref | Ref | Ref | Ref |

Note: Fixed effect models use only the within-child variance and the estimate is an average of the estimated association between a change in poverty status, or a change in income on a change in the outcome for each included child. The number included in the fixed effect model is therefore small, as there must be a change in the outcome. Fixed effect models were adjusted for age, age squared and wave identifier. Age squared was not included in the Otitis Media models as it was not a significant predictor when tested. Random effect models were adjusted for age, age squared, and ethnicity. Note that an OR>1 for 'BHC income poverty' indicates greater poverty is associated with increased odds of hospitalisation, whereas an OR>1 for 'Log of BHC income' indicates greater income is associated with increased odds of hospitalisation. We conducted a sensitivity analysis using age groups–it made little difference. While we only present the results for BHC income, we did run these models for disposable and IRD income. The substantive findings were similar for all income types.

**Table 6. Unadjusted associations between measures of deprivation and hospitalisations in the SoFIE and Census samples.**

| | | Otitis Media | | Oral Health | | Infectious | | Respiratory | | Preventable | | Hosp admission | |
|---|---|---|---|---|---|---|---|---|---|---|---|---|---|
| **NZDep–SoFIE** | BHC–self report (%) | % | RR | % | RR | % | RR | % | RR | % | RR | % | RR |
| quintile 1 (least deprived) | 14.5 | 0.40 | 1.00 | 0.40 | 1.00 | 2.49 | 1.00 | 0.78 | 1.00 | 2.28 | 1.00 | 6.28 | 1.00 |
| quintile 2 | 19.9 | 0.37 | 0.93 | 0.41 | 1.03 | 2.49 | 1.00 | 0.91 | 1.18 | 3.09 | 1.35 | 7.21 | 1.15 |
| quintile 3 | 26.9 | 0.63 | 1.59 | 0.49 | 1.25 | 2.87 | 1.15 | 1.48 | 1.91 | 3.81 | 1.67 | 7.86 | 1.25 |
| quintile 4 | 32.9 | 0.78 | 1.98 | 0.39 | 0.99 | 3.25 | 1.31 | 1.05 | 1.35 | 3.49 | 1.53 | 7.50 | 1.19 |
| quintile 5 (most deprived) | 53.9 | 0.87 | 2.20 | 0.70 | 1.76 | 4.18 | 1.68 | 1.25 | 1.61 | 4.15 | 1.82 | 7.95 | 1.27 |
| **NZDep–Census** | BHC–self report (%) | % | RR | % | RR | % | RR | % | RR | % | RR | % | RR |
| quintile 1 (least deprived) | 7.6 | 0.31 | 1.00 | 0.36 | 1.00 | 2.15 | 1.00 | 0.62 | 1.00 | 2.06 | 1.00 | 6.03 | 1.00 |
| quintile 2 | 14.6 | 0.38 | 1.22 | 0.41 | 1.15 | 2.46 | 1.14 | 0.77 | 1.24 | 2.43 | 1.18 | 6.61 | 1.10 |
| quintile 3 | 21.0 | 0.44 | 1.44 | 0.53 | 1.48 | 2.84 | 1.32 | 0.84 | 1.35 | 2.80 | 1.36 | 7.16 | 1.19 |
| quintile 4 | 29.8 | 0.47 | 1.53 | 0.66 | 1.84 | 3.22 | 1.50 | 1.02 | 1.65 | 3.24 | 1.57 | 7.70 | 1.28 |
| quintile 5 (most deprived) | 47.6 | 0.51 | 1.66 | 0.83 | 2.32 | 3.79 | 1.76 | 1.32 | 2.13 | 3.93 | 1.90 | 8.20 | 1.36 |
| **Material Deprivation–SoFIE** | BHC–self report (%) | % | RR | % | RR | % | RR | % | RR | % | RR | % | RR |
| None | 16.9 | 0.33 | 1.00 | 0.29 | 1.00 | 2.33 | 1.00 | 0.75 | 1.00 | 2.53 | 1.00 | 6.37 | 1.00 |
| One | 32.8 | 0.57 | 1.75 | 0.46 | 1.60 | 3.09 | 1.33 | 0.86 | 1.14 | 2.49 | 0.98 | 6.52 | 1.02 |
| Two | 43.5 | 0.85 | 2.60 | 0.64 | 2.23 | 3.60 | 1.55 | 0.76 | 1.02 | 2.88 | 1.14 | 6.92 | 1.09 |
| Three/four | 56.5 | 0.87 | 2.67 | 0.74 | 2.58 | 3.48 | 1.50 | 1.02 | 1.36 | 3.33 | 1.32 | 7.60 | 1.19 |
| Five or more | 75.4 | 0.83 | 2.55 | ¥ | | 3.33 | 1.43 | 1.32 | 1.76 | 4.31 | 1.70 | 8.88 | 1.39 |

¥ There were too few children who were hospitalised for oral health conditions and had 5 or more indicators of material deprivation, therefore for this comparison we restricted the material deprivation to none, one, two, three or more.

for the Census. A similar pattern of findings was found for material deprivation (SoFIE data only). Children experiencing five or more indicators of deprivation had risks of hospitalisation that were 1.39–2.55 times greater than children who experienced no deprivation.

We tested two additional sensitivity analyses using SoFIE data. First, to test whether attrition impacted the associations between income and hospitalisations we estimated associations between income poverty and hospitalisations by survey wave. There was no evidence for different magnitudes of associations in earlier waves (which were less affected by attrition) than later waves (see S1 Table). Second, to account for the error associated with single measures of income, we considered the association between income averaged across two previous waves with hospitalisations. Wave averaged income, and the log of wave averaged income also showed a non-linear relationship with hospitalisation, therefore we used quintiles of income. For most outcomes the bottom three averaged-income quintiles showed a similar risk for hospitalisation to each other, but elevated risk compared to the highest 20% of the income distribution (see S2 Table).

## Discussion

Income poverty is weakly associated with children's health outcomes in New Zealand. This was found in two cohorts, a longitudinal panel and cross-sectional Census cohorts, and persisted when considering cumulative effects, controlling for unmeasured confounding in fixed effects models, and controlling for random effect models adjusted for age and ethnicity. However, income poverty was associated with other measures of deprivation: those in income poverty reported higher levels of material deprivation and were more likely to reside in deprived areas (similar magnitudes of associations were found in SoFIE and the census). Further, these non-income measures of deprivation were more strongly associated with hospitalisation outcomes, typically in a dose-response fashion, and associations were stronger than associations found using measures of income poverty.

These findings add to the inconsistent results for health outcomes presented by Cooper and Stewart's (2017) systematic review [16]. Whereas we report at best weak associations for range of hospitalisation outcomes, including for respiratory conditions, Cooper and Stewart (2017) report no effect of income poverty on asthma, wheezing and other respiratory diseases in childhood (albeit, based on only three studies). As with the current study, evidence from UK studies suggests that the association between income and child health is weak, and in many cases a direct effect of income does not exist [37–41]. There is also a suggestion that associations may be weaker for 'objective' measures of health (as measured here) compared to subjective measures [40].

Our findings for income directly contrast with the overwhelming evidence demonstrating the importance of socioeconomic conditions more broadly for children's health, cognitive and social development [16, 42–44], and the long-term impacts this has for adult health [43, 45]. Income is often the subject of study because it is the easiest to manipulate through policy measures such as income redistribution measures. Yet, it is notoriously difficult to obtain accurate estimates of the disposable income families or households have available to them to spend on goods and services [46]. Measures of income struggle to capture the complex and interacting systems of financial support, wealth accumulation, assets, and financial liabilities [46]. A long window of follow up would be needed to assess economic stability and returns on investments. Therefore, two people with the same income estimate over a shorter time period, may be managing, spending and investing that money in very different ways, with differential impacts on their children's immediate and long-term health outcomes. These differences may be captured

by other socioeconomic aspects such as parental education and tenure, but not by income per se.

Material deprivation may provide a more reliable measure of poverty as it measures access to basic resources and services such as sufficient food and heating, and ability to cope with an unexpected expense [3]. Income as a sole measure of poverty may be too simplistic. There is a growing interest in multidimensional poverty [47–50]. Within this framework, low income would be one of many indicators of poverty, alongside material deprivation, and would not be a sufficient indicator of poverty by itself. By extension, alleviation of poverty and reducing its impact on health may require solutions that do not focus solely on income, but also target other indicators of poverty such as material deprivation.

The results of this study should be considered alongside the following limitations. First, our health outcomes are limited to hospitalisations. 12-month hospitalisation rates among children are low, especially hospitalisations for Otitis Media and Oral Health. We note also that our hospitalisation measures are measures of health care utilisation–we did not directly measure illness or health status so cannot draw conclusions about the nature of the association between poverty and these outcomes. There may be a service use bias in who gets hospitalised, whereby sick children from low income families may be less likely to go to hospital [51]. However, if this were the case, we would not expect to see the dose-response type relationships we see by area level deprivation, and associations with material deprivation.

Second, other health outcomes were not able to be linked to either of the cohorts we analysed. For example, no primary health care (general practitioner) data are available for analysis. Primary care visits may be a more sensitive measure of childhood illness than hospitalisation, but even if available, this outcome measure could potentially bias the analysis because transport costs and other resource limitations may prevent families in poverty from accessing primary care (primary care visits for children incurred a cost during the study period, 2002–2010). Hospitalisation, although itself potentially subject to mismeasurement from differential service use, provides a more consistent threshold for service access, and is free in New Zealand.

Third, material deprivation was only asked as part of the health module at waves 3, 5 and 7 of SoFIE, limiting the ability to investigate hospitalisations in relation to material deprivation. More in depth longitudinal data sources are required to investigate a fuller range of health outcomes in relation to income poverty, material deprivation and multidimensional poverty.

## Conclusions

We find weak evidence for an association between income poverty and hospitalisation in childhood and no evidence of a causal relationship. This suggests that income poverty has no causal effect on children's hospitalisations, at least in the short run (up to eight years). Other measures of deprivation showed demonstrably larger associations with the same hospitalisation outcomes. While the literature clearly demonstrates that poverty is important for child health, our study findings suggest that income measures alone may not be sufficient to capture the diversity of household economic circumstances.

### Statistics New Zealand disclaimer

The results in this paper are not official statistics. They have been created for research purposes from the Integrated Data Infrastructure (IDI), managed by Statistics New Zealand. The opinions, findings, recommendations, and conclusions expressed in this paper are those of the authors, not Statistics NZ. Access to the anonymised data used in this study was provided by Statistics NZ under the security and confidentiality provisions of the Statistics Act 1975. Only

people authorised by the Statistics Act 1975 are allowed to see data about a particular person, household, business, or organisation, and the results in this paper have been confidentialised to protect these groups from identification and to keep their data safe. Careful consideration has been given to the privacy, security, and confidentiality issues associated with using administrative and survey data in the IDI. Further detail can be found in the Privacy impact assessment for the Integrated Data Infrastructure available from www.stats.govt.nz.

The results are based in part on tax data supplied by Inland Revenue to Statistics NZ under the Tax Administration Act 1994. This tax data must be used only for statistical purposes, and no individual information may be published or disclosed in any other form, or provided to Inland Revenue for administrative or regulatory purposes. Any person who has had access to the unit record data has certified that they have been shown, have read, and have understood section 81 of the Tax Administration Act 1994, which relates to secrecy. Any discussion of data limitations or weaknesses is in the context of using the IDI for statistical purposes, and is not related to the data's ability to support Inland Revenue's core operational requirements.

## Supporting information

**S1 Appendix. International statistical classification of diseases and related health problems 10th revision (ICD-10) codes used for hospitalisations.**
(DOCX)

**S2 Appendix. Association between household equivalised income and hospitalisations using local polynomial regressions.**
(DOCX)

**S1 Table. Association between income poverty and hospitalisations stratified by wave for SoFIE children.**
(DOCX)

**S2 Table. Association between quintiles of BHC self-reported income in SoFIE averaged over two waves with hospitalisations.**
(DOCX)

## Acknowledgments

We acknowledge Tony Blakely for helpful comments on the manuscript; Kristie Carter, Fiona Imlach and Trinh Le for their help with understanding the SoFIE dataset; and Stats NZ for provided data access and undertaking confidentiality checks on outputs.

## Author Contributions

**Conceptualization:** Nichola Shackleton, Barry J. Milne.

**Data curation:** Nichola Shackleton, Barry J. Milne.

**Formal analysis:** Nichola Shackleton, Eileen Li, Sheree Gibb, Barry J. Milne.

**Funding acquisition:** Nichola Shackleton, Sheree Gibb, Amanda Kvalsvig, Michael Baker, Andrew Sporle, Rebecca Bentley, Barry J. Milne.

**Methodology:** Nichola Shackleton, Barry J. Milne.

**Writing – original draft:** Nichola Shackleton, Barry J. Milne.

**Writing – review & editing:** Nichola Shackleton, Eileen Li, Sheree Gibb, Amanda Kvalsvig, Michael Baker, Andrew Sporle, Rebecca Bentley, Barry J. Milne.

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
