## [Decision Letter · Decision Letter 0]

30 Sep 2020

PONE-D-20-23834

The relationship between income poverty and child hospitalisations in New Zealand: evidence from longitudinal household panel data and Census data

PLOS ONE

Dear Dr. Milne,

Thank you for submitting your manuscript to PLOS ONE. We obtained only one review, which states that the manuscript has potential and recommends a major revision. We agree with this reviewer invite you to submit a revised version of the manuscript that addresses the points raised during the review process.

The reviewer comments are below. In my opinion, the main concern is that the manuscript must address the endogeneity of income poverty. The current version presents evidence of the association between poverty and hospitalizations. Also, the manuscript should explore the mechanisms by which poverty affects hospitalizations.

We look forward to receiving your revised manuscript.

Kind regards,

Gabriel A. Picone

Academic Editor

PLOS ONE

Journal Requirements:

2.Thank you for including your ethics statement:  "This study received ethics approval from the University of Auckland Human Ethics Committee (019618). Consent to undertake the research and to link to administrative data sources was provided by the parents of SoFIE participants. Access to the anonymised data used in this study, including both SoFIE and Census data, was provided by Statistics NZ under the security and confidentiality provisions of the Statistics Act 1975 (see below).

Statistics New Zealand Disclaimer

The results in this paper are not official statistics. They have been created for research purposes from the Integrated Data Infrastructure (IDI), managed by Statistics New Zealand. The opinions, findings, recommendations, and conclusions expressed in this paper are those of the authors, not Statistics NZ. Access to the anonymised data used in this study was provided by Statistics NZ under the security and confidentiality provisions of the Statistics Act 1975. Only people authorised by the Statistics Act 1975 are allowed to see data about a particular person, household, business, or organisation, and the results in this paper have been confidentialised to protect these groups from identification and to keep their data safe. Careful consideration has been given to the privacy, security, and confidentiality issues associated with using administrative and survey data in the IDI. Further detail can be found in the Privacy impact assessment for the Integrated Data Infrastructure available from www.stats.govt.nz.

The results are based in part on tax data supplied by Inland Revenue to Statistics NZ under the Tax Administration Act 1994. This tax data must be used only for statistical purposes, and no individual information may be published or disclosed in any other form, or provided to Inland Revenue for administrative or regulatory purposes. Any person who has had access to the unit record data has certified that they have been shown, have read, and have understood section 81 of the Tax Administration Act 1994, which relates to secrecy. Any discussion of data limitations or weaknesses is in the context of using the IDI for statistical purposes, and is not related to the data’s ability to support Inland Revenue’s core operational requirements.".   

Please provide additional details regarding participant consent. In the ethics statement in the Methods and online submission information, please ensure that you have specified what type you obtained (for instance, written or verbal, and if verbal, how it was documented and witnessed). If your study included minors, state whether you obtained consent from parents or guardians. If the need for consent was waived by the ethics committee, please include this information.

3.We note that you have indicated that data from this study are available upon request. PLOS only allows data to be available upon request if there are legal or ethical restrictions on sharing data publicly. For information on unacceptable data access restrictions, please see http://journals.plos.org/plosone/s/data-availability#loc-unacceptable-data-access-restrictions.

Reviewers' comments:

Reviewer's Responses to Questions

**Comments to the Author**

1. Is the manuscript technically sound, and do the data support the conclusions?

Reviewer #1: Yes

2. Has the statistical analysis been performed appropriately and rigorously? 

Reviewer #1: Yes

3. Have the authors made all data underlying the findings in their manuscript fully available?

Reviewer #1: No

4. Is the manuscript presented in an intelligible fashion and written in standard English?

Reviewer #1: Yes

5. Review Comments to the Author

Reviewer #1: The study uses data from New Zealand to investigate the association between income and hospitalizations among children age 0-17. The authors find little evidence that poverty is associated with higher hospitalization rates among children when defining poverty as being below the median income of 60%. When using measures of deprivation instead, the analysis provides suggestive evidence for slightly stronger association with hospitalizations.

The research questions of the study are very important and the authors conduct a through empirical analysis to provide evidence answering these questions. Having said this, I believe there are a few parts missing in the paper that should be added and clarified:

1) In the introduction (from line 64), the authors mention that so far only very little evidence exists on the causal relationship between poverty and childhood health. Based on my interpretation of the study setup and the data used, I do not see how this paper is contributing to this. While the analysis uses high quality data and established research methods, the results only provide evidence for associations. As mentioned by the authors, this has been done by many others in the past. Thus, the authors need to make a better case for how their study contributes to the existing literature. Is using data from New Zealand one of the contributions or are there previous studies analysis the questions at hand with New Zealand data? I would recommend having a subsections highlighting the contributions of the study, especially given that these might be small overall.

From my understanding, previous studies have provided causal estimates by using exogenous variations in income as a result of changes in income assistance programs, for example.

2) I believe the study would benefit from an upfront discussion on the potential pathways underlying the relationship between poverty (or deprivation in general) and child health, with a specific focus on the policy background in New Zealand. This should cover a discussion on access to quality health care as well as the affordability of such care. Other pathways, among many others, could be health behaviors and health education. Could it be that less wealthy parents are less aware of certain health risks and therefore are less likely to take their children to the hospital when they suffer certain conditions? Can they not afford to pay for certain treatments? In addition to including a section discussing potential mechanisms, I would recommend including directly testing for the role of some of these in the analysis if feasible in the data. This would help with comment 1 regarding the contributions of the study and could help tell a complete story explaining the (lack of) associations found in the main analysis.

3) I agree that the use of longitudinal data provides benefits for the research question at hand. The paper mentions that the FE analysis are restricted to the subsample who have changes in recorded hospitalizations per year. In my understanding, you would also need variations in income/poverty status over time in this analysis. Can you provide evidence for the share of households in your sample that changes poverty status over time? If this share is pretty low, I believe this could explain the lack of significant association in the analysis. More information should be provided to increase the credibility of this model?

6. PLOS authors have the option to publish the peer review history of their article (what does this mean?). If published, this will include your full peer review and any attached files.

Reviewer #1: No

---

## [Author Response · Author response to Decision Letter 0]

13 Nov 2020

**Note, these are also contained in the "response to reviewers" document, which also contains additional analyses requested.

Associate Editor comments

In my opinion, the main concern is that the manuscript must address the endogeneity of income poverty.

This has been addressed as described in our response to comment 1 of reviewer 1 below, and is described on p4-6 of the revised manuscript.

Also, the manuscript should explore the mechanisms by which poverty affects hospitalizations.

This has been addressed as described in our response to comment 2 of reviewer 1 below, and in the additional analyses we describe and present at the end of this document. We also describe these mechanisms on p3-4 of the revised manuscript.

We believe this has been done.

2. Please provide additional details regarding participant consent. In the ethics statement in the Methods and online submission information, please ensure that you have specified what type you obtained (for instance, written or verbal, and if verbal, how it was documented and witnessed). If your study included minors, state whether you obtained consent from parents or guardians. 

We have amended the ethics statement in the submission form, and in the methods section (lines 142-145):

“This study received ethics approval from the University of Auckland Human Ethics Committee (019618). Access to the anonymised data used in this study, including both SoFIE and Census data, was provided by Statistics NZ under the security and confidentiality provisions of the Statistics Act 1975.”

Also lines 153-156:

“Written consent was obtained from 79% adult participants to link to administrative health records21. Children were excluded from analyses if they had a consent-refusing parent at any wave, or if they refused consent themselves when aged 15 or over.”

Also, the Statistics New Zealand disclaimer regarding use of the data has been included after the acknowledgements, lines 532-552:

“Statistics New Zealand Disclaimer

The results in this paper are not official statistics. They have been created for research purposes from the Integrated Data Infrastructure (IDI), managed by Statistics New Zealand. The opinions, findings, recommendations, and conclusions expressed in this paper are those of the authors, not Statistics NZ. Access to the anonymised data used in this study was provided by Statistics NZ under the security and confidentiality provisions of the Statistics Act 1975. Only people authorised by the Statistics Act 1975 are allowed to see data about a particular person, household, business, or organisation, and the results in this paper have been confidentialised to protect these groups from identification and to keep their data safe. Careful consideration has been given to the privacy, security, and confidentiality issues associated with using administrative and survey data in the IDI. Further detail can be found in the Privacy impact assessment for the Integrated Data Infrastructure available from www.stats.govt.nz.

The results are based in part on tax data supplied by Inland Revenue to Statistics NZ under the Tax Administration Act 1994. This tax data must be used only for statistical purposes, and no individual information may be published or disclosed in any other form, or provided to Inland Revenue for administrative or regulatory purposes. Any person who has had access to the unit record data has certified that they have been shown, have read, and have understood section 81 of the Tax Administration Act 1994, which relates to secrecy. Any discussion of data limitations or weaknesses is in the context of using the IDI for statistical purposes, and is not related to the data’s ability to support Inland Revenue’s core operational requirements.”

3.We note that you have indicated that data from this study are available upon request. PLOS only allows data to be available upon request if there are legal or ethical restrictions on sharing data publicly… a) If there are ethical or legal restrictions on sharing a de-identified data set, please explain them in detail (e.g., data contain potentially identifying or sensitive patient information) and who has imposed them (e.g., an ethics committee). Please also provide contact information for a data access committee, ethics committee, or other institutional body to which data requests may be sent.

This is described in more detail below, and on the submission form. We also include text suggested by the editorial team to confirm that other researchers may apply for access to the dataset used in our study via Statistics New Zealand.

“Integrated Data Infrastructure (IDI) data cannot be shared publicly because of the security and confidentiality provisions of the Statistics Act 1975 (NZ). Data are held on servers maintained by Statistics New Zealand and are never released to researchers or posted on other sites. As such, the dataset is not owned or available to be distributed by the authors.

However, the authors had no special access privileges and other researchers are able to access the data in the same manner as the authors. Researchers who wish to access the data reported in the manuscript for analysis (or access IDI data generally) must submit an application through Statistics New Zealand (https://www.stats.govt.nz/integrated-data/integrated-data-infrastructure#how-apply). Data requests and queries about data access may be sent to access2microdata@stats.govt.nz. 

If approved, data access is provided through a secure ‘Data Lab’ environment: a protected virtual environment in secure research facilities on computers that can access the IDI server, but nothing else (i.e. computer hard and soft drives cannot be accessed, and there is no access to the worldwide web). As such, IDI data are never sent to researchers or made available on data archives or via the worldwide web, but instead access is granted to analyse data within the Data Lab environment. Only the results of analyses (e.g., tables, models) can be requested to be released, and these must be confidentialised (e.g., all frequency counts are ‘random-rounded’ to be divisable by 3). 

More details on IDI data access is provided in reference 18 of the manuscript (Milne et al., 2019) and here: https://www.stats.govt.nz/integrated-data/integrated-data-infrastructure#data-safe.”

Reviewer 1 comments

1) In the introduction (from line 64), the authors mention that so far only very little evidence exists on the causal relationship between poverty and childhood health. Based on my interpretation of the study setup and the data used, I do not see how this paper is contributing to this. While the analysis uses high quality data and established research methods, the results only provide evidence for associations. As mentioned by the authors, this has been done by many others in the past. Thus, the authors need to make a better case for how their study contributes to the existing literature. Is using data from New Zealand one of the contributions or are there previous studies analysis the questions at hand with New Zealand data? I would recommend having a subsections highlighting the contributions of the study, especially given that these might be small overall.

From my understanding, previous studies have provided causal estimates by using exogenous variations in income as a result of changes in income assistance programs, for example.

This paper contributes to the evidence base on the causal relationship between poverty and childhood health through our fixed-effects analyses (Table 5), which are considered by many to identify causal associations (under certain assumptions) by isolating whether within individual change in poverty status/income is associated with change in health (hospitalisations in our case). The systematic reviews we cite include only studies which “use methods that allow us to reach conclusions about causal relationships” (Cooper & Stewart, 2013; 2017), and specifically include studies with ‘fixed effects’ design as we use, as well as studies that use exogenous variations in income (as the reviewer highlights). In response to the reviewer we clarify our use of the word causal in the introduction on lines 86-90:

“Cooper and Stewart (2013; 2017)10 16 synthesised evidence from studies that attempt to isolate the causal effect of low income on child outcomes, either through quasi-experimental designs (e.g., programmes that increased incomes) or by statistical isolating the impact of low-income by analysing within household differences or within-individual change.”

We also describe in more detail the nature of the causal associations found – and not found in the Cooper and Stewart (2017) review on lines 94-100:

“There was evidence pointing to significant effects of exogenous changes in income (i.e., through payments or tax credits) on birthweight and other neonatal outcomes, as well as evidence that differences in household income experienced by children within families explained birthweight differences between children. However, there was mixed evidence for the impact of exogenous changes in income on obesity and general health in later childhood, and no evidence (from three studies) for an effect of exogenous changes in income on asthma, wheezing and other respiratory diseases16”

As suggested by the reviewer we also include a subsection at the end of the introduction (lines 105-122) which highlights the contribution of our study:

“This study contributes to the existing literature in the following ways. First, it represents an analysis of associations between poverty / low income and a range of hospitalisations thought to be poverty sensitive, specifically: respiratory conditions19 20, infectious diseases8, otitis media18, oral health21 22, and preventable admissions23 24. Second, we assess poverty in two samples: a longitudinal household panel survey: the Survey of Family, Income and Employment25; and whole population data from the 2013 New Zealand census; and we measure income from two sources: self-report and administrative tax records. Third, there have been debates regarding ‘sensitive periods’ for poverty experience, with some emphasising the importance of early childhood,26 while others pointing to the importance of adolescence.27 We make use of our multi-age cohorts to assess whether associations between poverty and health are age-sensitive. Fourth, we make use of longitudinal data to assess whether duration of time in poverty has an impact on hospitalisation outcomes, as has been suggested for other health outcomes.27 Fifth, our use of longitudinal data allows us to analyse within-individual change over an extended period - eight annual data collection waves - to estimate if associations are likely to be causal. Specifically, fixed effects regression analyses focus on within-individual change and control all ‘fixed’ differences (i.e., unchanging – or time invariant – confounding) between individuals that might otherwise explain associations between poverty and health.16 28”

2) I believe the study would benefit from an upfront discussion on the potential pathways underlying the relationship between poverty (or deprivation in general) and child health, with a specific focus on the policy background in New Zealand. This should cover a discussion on access to quality health care as well as the affordability of such care. Other pathways, among many others, could be health behaviors and health education. Could it be that less wealthy parents are less aware of certain health risks and therefore are less likely to take their children to the hospital when they suffer certain conditions? Can they not afford to pay for certain treatments? In addition to including a section discussing potential mechanisms, I would recommend including directly testing for the role of some of these in the analysis if feasible in the data. This would help with comment 1 regarding the contributions of the study and could help tell a complete story explaining the (lack of) associations found in the main analysis.

We thank the reviewer for this comment. We conceived of causal pathways in the design of this study but chose not to investigate them given there was little evidence of an association between poverty and the health outcomes we investigated. Specifically, we conceived of two potential pathways from poverty to health: lack of ‘economic investments’, and increased ‘family stress’. Taking on board the reviewers suggestion, we have tested these factors (described and shown at the end of this document). The results indicate little evidence that any of four factors – household crowding and food insecurity (as measures of economic investment) and parental smoking and parental psychological distress (as measures of family stress) – changed the associations between poverty and health. Moreover, only one factor – food insecurity – had any influence on the health outcomes we assessed. As such, and because describing the rationale, methods, results and discussing these findings takes up a lot of space, we do not believe that the addition of these results to the paper is justified. However, we are prepared to be guided by the editor and reviewer on this. 

We have added the following paragraph to the introduction (lines 62-75) as an ‘upfront discussion on the potential pathways underlying the relationship between poverty (or deprivation in general) and child health’:

“A number of theories have been posited to explain the relationship between poverty and child health. One theory – the ‘economic investment’ theory – suggests that more affluent parents are better able to ‘invest’ more in their children’s development (e.g., through investments in nutrition, clothing, learning resources, housing, and healthcare) and this leads to better outcomes.10 Affluence can also overcome issues of access to health care, particularly if cost is a barrier, but even when it is not: e.g., New Zealand has universal health care with low co-payments in primary care, yet the odds of unmet need for primary health care is 1.4 times greater among those living in the most versus least deprived neighbourhoods.11 Another theory – the ‘family stress’ theory – suggests that living in poverty increases parental stress levels, and this stress hinders parents’ ability to provide quality care, which has negative impacts upon children’s outcomes.12 13 Other factors, such as health behaviours and health literacy may also play a role.14 15 Note that these factors are not mutually exclusive: the effect of poverty on children’s health could operate simultaneously through many pathways.”

3) I agree that the use of longitudinal data provides benefits for the research question at hand. The paper mentions that the FE analysis are restricted to the subsample who have changes in recorded hospitalizations per year. In my understanding, you would also need variations in income/poverty status over time in this analysis. Can you provide evidence for the share of households in your sample that changes poverty status over time? If this share is pretty low, I believe this could explain the lack of significant association in the analysis. More information should be provided to increase the credibility of this model?

Overall, 28.1% of households change poverty status at least once during the eight longitudinal waves so it was relatively common. We acknowledge this on p13, lines 295-297:

“Note, poverty status was variable across waves: 28.1% of children changed poverty status at least once across the eight waves.”

Other changes made

A few errors were noticed with Table 5 when putting together analyses for this response. First, a number of confidence intervals were calculated incorrectly. In most cases this was because +-1.96 multiplied by the standard error had been applied directly to the odds ratio, rather than being applied to the unexponentiated logit coefficients before exponentiating to give confidence intervals on the odds scale. These have now been calculated correctly and the Table and preceding text has been updated (line 384). Second, the coefficients for “Log of BHC income” were incorrectly labelled as β when they are odds ratios. As all coefficients in the Table are odds ratios, all within-table labels have been removed and “Panels show odd ratios (95% confidence intervals)” has been added to the Table legend. Third, the results for the quintiles of BHC income had been calculated using an incorrect variable. The table has now been updated with the correct results.

Also, Tony Blakely has asked to be removed as an author and to be acknowledged only. This has been updated in the manuscript and on the submission system (a change of authorship form has been included as part of the submissions).

Thank you for considering our revised manuscript.

Barry Milne, on behalf of all authors.

---

## [Decision Letter · Decision Letter 1]

1 Dec 2020

The relationship between income poverty and child hospitalisations in New Zealand: evidence from longitudinal household panel data and Census data

PONE-D-20-23834R1

Dear Dr. Milne,

We’re pleased to inform you that your manuscript has been judged scientifically suitable for publication and will be formally accepted for publication once it meets all outstanding technical requirements.

Kind regards,

Gabriel A. Picone

Academic Editor

PLOS ONE

Additional Editor Comments (optional):

Reviewers' comments:

Reviewer's Responses to Questions

**Comments to the Author**

1. If the authors have adequately addressed your comments raised in a previous round of review and you feel that this manuscript is now acceptable for publication, you may indicate that here to bypass the “Comments to the Author” section, enter your conflict of interest statement in the “Confidential to Editor” section, and submit your "Accept" recommendation.

Reviewer #1: All comments have been addressed

2. Is the manuscript technically sound, and do the data support the conclusions?

Reviewer #1: Yes

3. Has the statistical analysis been performed appropriately and rigorously? 

Reviewer #1: Yes

4. Have the authors made all data underlying the findings in their manuscript fully available?

Reviewer #1: Yes

5. Is the manuscript presented in an intelligible fashion and written in standard English?

Reviewer #1: Yes

6. Review Comments to the Author

Reviewer #1: I would like to thank the authors for addressing my comments and suggestions in the revised manuscript. I have no further comments at this stage.

7. PLOS authors have the option to publish the peer review history of their article (what does this mean?). If published, this will include your full peer review and any attached files.

Reviewer #1: **Yes: **Otto Lenhart

---

## [Editor Report · Acceptance letter]

15 Dec 2020

PONE-D-20-23834R1 

The relationship between income poverty and child hospitalisations in New Zealand: evidence from longitudinal household panel data and Census data 

Dear Dr. Milne:

I'm pleased to inform you that your manuscript has been deemed suitable for publication in PLOS ONE. Congratulations! Your manuscript is now with our production department. 

Kind regards, 

on behalf of

Dr. Gabriel A. Picone 

Academic Editor

PLOS ONE